# Smad7 Binds Differently to Individual and Tandem WW3 and WW4 Domains of WWP2 Ubiquitin Ligase Isoforms

**DOI:** 10.3390/ijms20194682

**Published:** 2019-09-21

**Authors:** Lloyd C. Wahl, Jessica E. Watt, Hiu T. T. Yim, Danielle De Bourcier, James Tolchard, Surinder M. Soond, Tharin M. A. Blumenschein, Andrew Chantry

**Affiliations:** 1School of Biological Sciences, University of East Anglia, Norwich NR4 7TJ, UK; lloyd.wahl@cityu.edu.hk (L.C.W.); Jessica.Watt@newcastle.ac.uk (J.E.W.); tiffany.yim228@yahoo.co.uk (H.T.T.Y.); surinder.soond@yandex.ru (S.M.S.); 2School of Chemistry, University of East Anglia, Norwich NR4 7TJ, UKtolchard@ebi.ac.uk (J.T.)

**Keywords:** TGFβ signalling, transforming growth factor beta, smad, smad7, E3 ubiquitin ligase, NEDD4, WW domain, protein interaction

## Abstract

WWP2 is an E3 ubiquitin ligase that differentially regulates the contextual tumour suppressor/progressor TGFβ signalling pathway by alternate isoform expression. WWP2 isoforms select signal transducer Smad2/3 or inhibitor Smad7 substrates for degradation through different compositions of protein–protein interaction WW domains. The WW4 domain-containing WWP2-C induces Smad7 turnover in vivo and positively regulates the metastatic epithelial–mesenchymal transition programme. This activity and the overexpression of these isoforms in human cancers make them candidates for therapeutic intervention. Here, we use NMR spectroscopy to solve the solution structure of the WWP2 WW4 domain and observe the binding characteristics of Smad7 substrate peptide. We also reveal that WW4 has an enhanced affinity for a Smad7 peptide phosphorylated at serine 206 adjacent to the PPxY motif. Using the same approach, we show that the WW3 domain also binds Smad7 and has significantly enhanced Smad7 binding affinity when expressed in tandem with the WW4 domain. Furthermore, and relevant to these biophysical findings, we present evidence for a novel WWP2 isoform (WWP2C-ΔHECT) comprising WW3–WW4 tandem domains and a truncated HECT domain that can inhibit TGFβ signalling pathway activity, providing a further layer of complexity and feedback to the WWP2 regulatory apparatus. Collectively, our data reveal a structural platform for Smad substrate selection by WWP2 isoform WW domains that may be significant in the context of WWP2 isoform switching linked to tumorigenesis.

## 1. Introduction

Polyubiquitination, a process in which E3 ligase activity earmarks substrates for degradation at the proteasome by the attachment of ubiquitin chains, is an essential step in protein regulation [1,2]. This targeted degradation is part of the ubiquitin/proteasome system (UPS) which controls protein levels across a wide variety of physiological pathways. Dysfunction of this process is implicated in the pathogenesis of a range of diseases, including cancers, neurodegenerative diseases, inflammatory disorders and many others [3]. Accordingly, components of the UPS are the targets of several approved drugs and candidate targets for novel therapeutic intervention [4,5]. The protein–protein interaction domains of E3 ligases are particularly appealing targets, since they confer specificity to the ubiquitin pathway through selective binding of target substrates and they offer the lowest likelihood of off-target effects. Targeting protein–protein interaction domains has historically been considered challenging, but new approaches have facilitated the design of several successful protein–protein interaction domain inhibitors [6,7,8], including for an E3 ligase [9].

WWP2 belongs to a subset of nine closely related HECT E3 ligases referred to as the NEDD4 family. Their defining characteristic is the presence of modular protein–protein interaction structural motifs called WW domains which are used to recruit substrates with short proline-rich motifs [10]. Despite these short binding motifs and their simple three-stranded antiparallel β-sheet secondary structure, WW domains confer specificity to the NEDD4 E3 ligases by differentiating between a broad range of target substrates [11]. These substrates are often linked to oncogenic pathways such as the tumour suppressor PTEN, the oncogene OCT4 (octamer-binding transcription factor 4), genome stability components, the Wnt pathway, the EGFR (epidermal growth factor receptor) pathway and the TGFβ (transforming growth factor-β) pathway [12]. These signalling pathways are often disrupted in human malignancies, and several NEDD4 family members, including NEDD4, NEDD4L, SMURF1, SMURF2, WWP1 and WWP2, also have altered expression or splicing in cancer [13].

This has led to a significant effort to characterise the regulatory role of NEDD4 family members. In particular, interactions with components of the TGFβ signalling pathway have been intensively studied. These components include Smad2 and Smad3, which are phosphorylated by serine/threonine kinase receptors at the cell surface and translocate to the nucleus. Here they recruit cofactors at genetic elements to induce transcriptional activation or suppression of a large number of unrelated and often contradictory genes, including Smad7, which inhibits the pathway to create a negative feedback loop. The genes regulated by TGFβ are central to cytostasis, proliferation, apoptosis, cell survival, cell adhesion and migration, stem cell pluripotency, differentiation and extracellular matrix reorganisation. As a result, this pathway plays an important role in development [14], immunity [15] and tissue homeostasis and repair [16,17]. The powerful gene programmes are harnessed in cancers [18] and misregulated in connective tissue, skeletal, inflammatory and immune diseases [19,20,21]. The defining characteristic of TGFβ signalling is its ability to sense the local environment and elicit diverse signalling outcomes based on the cellular context [22]. This is achieved by an exquisite regulation of each step in the pathway by an array of independently regulated binding partners, creating a complex biological algorithm. UPS input into this algorithm involves the interplay between ubiquitin ligases and deubiquitinating enzymes such as the ubiquitin-specific proteinases [23,24]. The NEDD4 family of E3 ubiquitin ligases have been found to interact with and cause the degradation of both the activating components (Smad2, Smad3 and TGFβ receptor-I) and the inhibitory component (Smad7). These E3 ligases create their own complex web of regulation that includes differential isoform-splicing and phospho-regulation [25,26,27] and, in so doing, they integrate the UPS, kinase pathway crosstalk and splicing programmes into the regulation of TGFβ signalling.

The nature of the interaction between WW domain and Smad substrate is critical to this regulatory activity and in defining a niche for each ligase. NEDD4L, for example, binds to the PPxY motifs in the linker region of TGFβ-activated Smad2 and Smad3 exclusively through its second WW domain, but only once they have been phosphorylated by CDK8/9 [25]. This phosphorylation occurs two residues N-terminal to the PPxY motif and directly affects the interaction between the NEDD4L WW2 domain and Smad2/3, enhancing their turnover through proteasome-mediated degradation [26]. On the other hand, SMURF1 binds Smad7 with high affinity, translocates to the cytoplasm and is recruited to TGFβ receptor I (TβR-I) via Smad7 interaction with the receptor. Here, TβR-I and Smad7 are polyubiquitinated, causing their degradation at the proteasome [28,29]. The SMURF1/Smad7 interaction is mediated by a synergistic action between SMURF1 tandem WW domains, a mechanism that is found amongst other tandem WW domain-containing proteins [30] and is also evident in its homolog SMURF2 [31,32]. This allows the interaction to be tuned by alternative transcript splicing, and the incorporation of a stretch of amino acids between these tandem domains diminishes the coordinated high-affinity binding of Smad7 [31].

Isoform-mediated regulation of the TGFβ signalling pathway is not a feature unique to SMURF1. WWP2 is found to be expressed as three distinct isoforms that arise from alternative splicing and alternative promoter activity, with the capacity to either positively or negatively regulate TGFβ signalling [27]. This ability to regulate both the signal transducers (Smad2/3) and the negative feedback loop (Smad7) in a discrete manner is interesting, given the highly contextual nature of TGFβ signalling outcomes [33]. The divergent activity of WWP2 isoforms is apparently allowed by the incorporation of different WW domains with different substrate preferences within their sequence. WWP2-FL contains the full complement of WW domains and is observed to bind Smad2/3 and Smad7 in vivo, the N-terminal isoform WWP2-N has preference for Smad2/3 but also plays a role in relinquishing WWP2 autoinhibition, while the C-terminal isoform WWP2-C has preference for Smad7 [34]. Indeed, this substrate selectivity has been shown to play a direct role in the pathology of cardiac fibrosis through the regulation of Smad2 by WWP2-N [35], while the overexpression of WWP2 isoforms in early and advanced disease stages in cancer cDNA panel arrays indicates their potential as targets for therapeutic intervention [36]. As a negative regulator of Smad7, WWP2-C propagates downstream TGFβ signalling gene programmes, including that of epithelial–mesenchymal transition (EMT). The critical role this process plays in the invasion and migration of tumours during metastasis makes this isoform of great interest in drug development.

High-resolution protein structures and binding site elucidation are required to inform rational drug design [7]. Some effort has been made to elucidate the structure of WWP2, with success in the crystallisation of the HECT domain [37] and a WW2-HECT construct [38], but with difficulty encountered in the expression and purification of individual WW domains [39]. We also found WWP2 and its domains difficult to express and purify, but here we have succeeded in improving the purification yield of WWP2 WW domains. We used our approach to solve the structure of the WW4 domain by nuclear magnetic resonance (NMR) spectroscopy. By measuring chemical shift changes by NMR, we observed the in vitro interaction between WW4 and Smad7 and uncovered a potential role for phospho-regulation of WWP2-mediated Smad7 degradation. We present evidence for a novel WWP2 isoform called WWP2C-ΔHECT which utilises a tandem WW3–WW4 domain pair to regulate the TGFβ pathway. We explored the role these tandem domains play in substrate recruitment by using NMR to probe their Smad7 binding affinities. We found that the affinity of WW3 for Smad7 is significantly enhanced in the presence of its tandem WW4 domain. These findings have implications in the context of substrate competition between WWP2 isoforms and add new insights into the WWP2 isoform-mediated negative feedback loop of the TGFβ pathway.

## 2. Results

### 2.1. WWP2 WW4 Adopts The Canonical WW Domain Fold

To solve the purification problems frequently encountered when attempting to purify WWP2 WW domains, we expressed WW4 with an N-terminal small solubility-enhancing B1 domain of streptococcal protein G (GB1) tag (Figure 1A). This produced a significantly enhanced yield and stability during purification. Attempts at cleaving WW4 from its solubility tag resulted in comprehensive precipitation. Since interaction between the two conjugated domains could disrupt the native conformation of WW4, T_1_, T_2_ and heteronuclear Nuclear Overhauser effect (NOE) relaxation experiments were performed to confirm that each domain behaves independently (Appendix A), from which it was deemed acceptable to proceed with the tag intact. A series of NMR spectroscopy experiments were used to assign 89% of atoms (available in the Biological Magnetic Resonance Data Bank (BMRB) [40] under accession code 34407). The unassigned atoms corresponded to three residues at the N-terminus, one serine within loop 1 of WW4, five prolines (for which there are no observable amide resonances), and some sidechain atoms throughout the sequence. A 20-model water-refined ensemble was produced from structural calculations (available in the Protein Data Bank (PDB) under accession code 6RSS). No inter-domain NOEs were observed between residues of the GB1 and WW4 domains, further indicating that the presence of the tag did not interfere with the WW domain fold. As a result, the linker between the two domains showed a high level of flexibility, giving a high backbone root-mean-square deviation (RMSD) value (8.9 Å) and rendering global structure validation unsuitable. Instead, validation was performed on the GB1 and WW4 domains separately, a summary of which is shown in Table 1. The GB1 domain holds the expected fold published elsewhere [41,42,43] (Appendix A). WW4 holds the typical three-stranded antiparallel β-sheet with a right-handed twist seen in other WW domains (Figure 1B and Appendix A). The fold is initiated by a type II β-turn between the linker and the first β-strand. The first and second β-strands are connected by a β-turn (Loop 1) which is stabilised by hydrogen bonding between the sidechain of 455Thr and the amide groups 457Glu and 458Val. The second and third β-strands are linked by a five-residue loop (Loop 2), where the carboxyl group of residue 464Asp forms hydrogen bonds with the amide groups of 467Thr, 468Arg and 469Thr. The third β-strand extends four residues until 472Phe, which substitutes the typically conserved C-terminal tryptophan found in other WW domains.

### 2.2. WWP2 WW4 Binds to A Smad7 and Phospho-Smad7 PPxY Motif

In previous experiments, we found that WWP2-mediated Smad7 degradation was significantly enhanced on stimulation with TGFβ [27]. One possible explanation for this is enhanced substrate/ligase affinity by post-translational modification of Smad7. Submission of the Smad7 amino acid sequence to the phosphorylation-site prediction servers NetPhos 3.1 [44,45] and GPS 3.0 [46] identified S206 in close proximity to the PPxY motif as a potential phosphorylation site (Figure 2A, Appendix A). We searched the literature for experimental evidence of S206 phosphorylation and found that in vivo expression of the C-terminal portion of Smad7 resulted in phosphorylation at S206 [47]. Since WW4 domain-containing WWP2-C has preference for Smad7 interaction and degradation [27], and because this domain shares a reasonable level of sequence similarity with Smad7-binding domains of the SMURF homologs (Appendix A), we used WW4 as a binding partner in titration experiments for short Smad7 and S206-phosphorylated Smad7 (pSmad7) ligands. Using NMR spectroscopy to detect the local molecular environment perturbation, we observed peak migration in a series of heteronuclear single quantum coherence experiments (HSQCs) in which the ligand concentration was increased (Figure 2B and Appendix A, and Figure 2C and Appendix A). This process is indicative of a fast-exchange binding event between WW4 and Smad7/pSmad7 and confirms in vivo observations of interactions between the WWP2 WW4 domain and Smad7. Dissociation constants (K_D_) were calculated from the binding curve. This revealed a 2.3x increase in the affinity of WW4 for pSmad7 (0.5 ± 0.07 mM) when compared to unphosphorylated Smad7 (1.19 ± 0.21 mM) (Figure 2D).

Since changes in peak shift are caused by local electromagnetic field perturbation upon interaction with the ligand, plotting the peak shift distances of each titration allowed us to identify the binding site (Figure 2E,F). This provided insight into how WWP2-C binds to Smad7 in vivo and identified two binding hotspot residues, 470Thr and 471Thr (in red). The pattern of binding across the WW4 domain was almost identical for both phosphorylated and unphosphorylated Smad7. There were differences in shift distance when comparing the same residues across the binding site, but no obvious variations that might indicate direct polar contacts with the phosphate group, as seen in other examples [26]. All three β-strands exhibited electromagnetic field perturbation on ligand binding, with the third β-strand exhibiting the most significant changes. Previous studies on WW domain/ligand co-structures and binding assays identified two characteristic binding pockets on the WW domain surface that determine specificity for the PPxY recognition motif. These are the XP-binding groove, which binds the proline-rich N-terminal region [48], and a second specificity pocket that accommodates the tyrosine [49]. In WW4, the XP-binding groove is formed by 461Tyr and 472Phe, while the second specificity pocket is formed by 463Val, 465His and 468Arg. Both pockets can be seen in the structure and both are involved in binding (Figure 2F).

### 2.3. Evidence for A Novel WWP2 Isoform Which Contains Tandem WW3–WW4 Domains and A Truncated HECT Domain That Can Inhibit TGFβ-Dependent Signalling Activity

In various western blots using an antibody specific for an epitope in the C-terminal region of WWP2, we noticed cross-reactivity with a small protein band at approximately 33 kDa (Figure 3A). Similar to other WWP2 isoforms, expression of this protein was inducible by TGFβ, and we were able to suppress its expression in epithelial cells by transfection of various epithelial splicing regulatory proteins (ESRPs) that are known to regulate specific gene splice events during the TGFβ-induced epithelial cell differention programme [50]. Since no other protein-coding transcript registered in the Genbank database could produce a protein product that was both the correct size and cross-reactive with the anti-WWP2C antibody, we suspected that this band belonged to a novel WWP2 isoform not yet reported in the literature. Searching the expressed sequence tag (EST) database produced two hits within intronic regions of the *wwp2* gene (Figure 3B). There is a TATA box within intron 9/10, 5′ of the DC341937.1 EST, indicating promoter activity that may control the expression of a novel isoform. This intron contains a cluster of growth factor-responsive elements (Appendix A), and we observed a transcriptional response to TGFβ and EGF stimulation in luciferase reporter assays (Appendix A ). We found evidence of *wwp2* transcript splice variation corresponding to both EST DC341937.1 (Figure 3C) and EST BX471495.1 (Figure 3D) in metastatic cell lines. The theoretical product of a transcript with both ESTs is a WWP2C-ΔHECT isoform of approximately 38 kDa that includes tandem WW3 and WW4 domains and a potentially catalytically inactive HECT domain (Figure 3E). We created a construct that corresponds to this novel isoform and co-transfected HEK293A cells with the Smad3-dependent CAGA12-luciferase reporter to test the effect of WWP2C-ΔHECT overexpression on TGFβ pathway transcriptional activity. We observed significant inhibition of Smad-dependent promoter activity in cells stimulated with TGFβ (Figure 3F). This suggests that, unlike WWP2-C which potentiates TGFβ signalling [27], the tentative WWP2C-ΔHECT isoform could suppress TGFβ/Smad-dependent signalling.

### 2.4. WWP2 WW Domain Affinity for Smad7 Is Switched On when Expressed in Tandem

The last tryptophan of WWP2 WW3 and the first tryptophan of WWP2 WW4 are separated by only 17 amino acids, and their role as tandem domains in both WWP2-FL and the tentative novel WWP2C-ΔHECT is undefined. WW domains are commonly found in other proteins as pairs, linked by a short stretch of amino acids, and are found to participate in cooperative or destructive binding in order to modulate substrate affinity. Since WWP2C-ΔHECT appears to inhibit TGFβ signalling despite lacking the cysteine required to catalyse the transfer of ubiquitin, we suspected that it may be interacting with the inhibitory Smad7 in an as yet undefined regulatory mechanism involving its tandem WW domains. To explore the role this tandem conformation plays in substrate recruitment, we used NMR titrations to probe the Smad7 affinity of tandem WW3–4, using a short PPxY Smad7 ligand recombinantly expressed in *Escherichia coli*. Due to purification requirements, this peptide is four additional amino acids longer at the N-terminus than the synthetic peptide previously used. We observed evidence of ligand binding in the WW3–4 Smad7 titration HSQCs (Figure 4A and Appendix A). Closer inspection of the binding pattern revealed that peaks belonging to WW4 exhibited fast-exchange migration, and so too did peaks belonging to the WW3 domain, some of which also exhibited intermediate exchange peak broadening. When we performed the same titrations on WW3 and WW4 as individual domains, we found that WW3 (Figure 4B and Appendix A), as well as WW4 (Figure 4C and Appendix A), bound Smad7 independently. For these single domain titrations, binding did not reach saturation, but curve fits to fast-exchange peak migration patterns were good, and errors were small enough to determine the dissociation constants. This revealed a greater affinity for WW3/Smad7 (139 ± 14.4 µM) compared to WW4/Smad7 (237 ± 15.7 µM). The observed affinity of WW4 for Smad7 was much higher for the expressed ligand compared to our synthetic peptide titrations in Figure 2B (1.19 ± 0.21 mM). This may be a result of the longer ligand holding a more favourable binding conformation or it could be due to synthetic peptide contaminants that disrupted binding, and indeed we found non-specific peak migration across GB1 and WW4 domains at high synthetic peptide concentrations.

Since the pattern of binding was similar for the respective domains in the single and tandem domain titrations (Figure 4D), it was assumed that WW3–4 binds the Smad7 ligand in a 2:1 ligand–receptor stoichiometry, with each WW domain binding site accepting one ligand. For the tandem domain data, the dissociation constants were calculated by fitting a curve to the combined binding data using a two-site binding equation (Figure 4E). Although this produced dissociation constants with higher standard deviations (Figure 4F), we were able to discern certain characteristics. Significantly, when both domains were expressed in tandem, WW3 domain affinity increased approximately seven-fold to 20.57 ± 26.24 µM, while WW4 affinity remained approximately the same at 249.3 ± 68.1 µM. These results are consistent with the raw data, which show WW3 residues reached close to saturation at lower ligand concentrations compared to WW4. The tandem WW4 domain peaks exhibited an identical, albeit consistently smaller, pattern of peak shift change as those of the single domain (Figure 4D), and the WW4 peaks of the free HSQC spectra for both constructs had identical chemical shifts. These observations strongly suggest that WW4 holds the same conformation and has the same binding mechanism in both single and tandem proteins, while the consistently smaller change in peak shift across the tandem WW4 binding site is expected of a domain in competition with its neighbour. The WW3 domain constructs showed distinctive differences in peak shift change. Tandem WW3 residues 414Arg and 420Val, in particular, had larger changes in shift (Figure 4D), while three additional tandem WW3 residues were in intermediate exchange between bound and free states (Appendix A). Further, comparison of the free HSQC amide peaks revealed differences in chemical shift across the entire domain (Figure 4G) and peaks that were missing in the tandem WW3 due to conformational exchange. This suggests that the WW3 conformation is influenced by the presence of WW4, sampling multiple conformations in the absence of Smad7. The increase in affinity for WW3 is in agreement with observations made of other tandem WW domains which often exhibit cooperation [30]. Unfortunately, this mechanism could not be explored structurally because of the conformational exchange in the WW3–4 construct. Instead, we were able to confirm the secondary structure of WW3, predicted from the backbone chemical shifts by the CSI 3.0 webserver [51]. This showed a three β-strand conformation typical of other WW domains, albeit with longer loop regions when compared to WW4 (Figure 4D and Appendix A). By mapping the titration data to the secondary structure, we found that all amide resonances belonging to the third WW3 β-strand bound Smad7 in intermediate exchange, both in tandem orientation and as a single domain. Unlike WW4, WW3 residues belonging to loop 1 were also involved in ligand coordination. This is a notable feature of other NEDD4 family WW domains with high Smad7 affinity [32,52].

## 3. Discussion

We presented here the previously unreported structure of the fourth WW domain of WWP2 and showed that it holds the typical three-stranded antiparallel β-fold found in other WW domains. The WWP2 WW4 domain binds Smad7 with residues across the three β-strands but has an unexpectedly low affinity. However, we found evidence for post-translational modification of Smad7 in close proximity to the PPxY motif that enhances WW4 affinity in vitro. Furthermore, we provided evidence for a novel TGFβ-inducible WWP2C-ΔHECT isoform that contains tandem WW3–WW4 domains and a potentially catalytically inactive HECT domain and which has the ability to reduce TGFβ/Smad-dependent gene expression. Notably, the WW3 and WW4 domains of WWP2 were shown to cooperate in Smad7 ligand coordination when expressed in tandem, significantly enhancing WW3 affinity and demonstrating the importance of tandem WW domains in substrate recruitment by WWP2 isoforms. This has implications on the study of WW domain binding affinities out of their native environment, as neighbouring domains have the capacity to dramatically alter binding affinity.

We previously showed that WWP2-C has a binding preference for Smad7 over Smad2/3 in vivo using co-immunoprecipitation experiments [27]. This appears to enhance Smad7 degradation via WWP2-C and increases Smad2/3-dependent gene expression. We therefore expected to observe high WW4/Smad7 affinity in vitro, but our results indicate only a moderate affinity. This affinity is lower than those of other NEDD4 WW domains, which have dissociation constants in the low micromolar range [32,52]—although determined using a different biophysical technique. Since inter-domain and domain-linker effects were found to be important in WW domain interactions both here and in other studies [30,38,53,54,55], we can speculate that a WW4 conformation more favourable to Smad7 binding is stabilised by the HECT domain or the linker region. However, studies on WW domains with low micromolar dissociation constants noted the involvement of extended recognition pockets where additional WW domain residues along the first β-strand and within loop 1 bind residues either side of the PPxY motif. For example, the Smad7 co-structure of the SMURF1 WW2 domain revealed two key interactions between 289Arg^SMURF1^/217D^Smad7^ C-terminal to the PPxY motif and 295Arg^SMURF1^/205E^Smad7^ N-terminal to the PPxY motif [32]. Mutational analysis of this domain found that exchanging 295Arg^SMURF1^ for glutamic acid reduced Smad7 affinity 12-fold, and indeed, WWP2 WW4 mirrors this substitution exactly, with the equivalent position occupied by 457Glu. This might confer some selectivity on WW4 towards substrates with basic residues proximal to their PPxY motifs. On the other hand, the WWP2 WW3 domain has both of these residues conserved (414Arg and 419Arg). Although the WW3 domain contains a deletion prior to 419Arg that might orient this residue on the opposite side of the β-fold, we observed substantial chemical shift change for both of these residues during Smad7 titration and a dissociation constant of comparable affinity, suggesting that the binding mechanism is similar.

The WW4 residue in the equivalent position of 289Arg^SMURF1^ is 453Lys. This might be expected to engage in a similar interaction, but in 15 out of 20 models of the structural ensemble, 453Lys forms a salt bridge with 451Glu and is most likely unavailable to coordinate Smad7 residues. This is reinforced by observations in Smad7 titration, which showed relatively minor changes in shift for 453Lys (Figure 2E). This is indicative of small changes in the chemical environment and unlikely to be due to direct ligand coordination. In four of these models, 451Glu also forms polar contacts with the second specificity pocket histidine. We propose that these interactions are important in NEDD4 WW domain ligand coordination, where Glu, at the +1 position relative to the first tryptophan, is 100% conserved, and either Lys or Arg occupy the equivalent position of 453Lys (+3) (Appendix A). In several solved structures [32,56,57,58,59,60,61], polar contacts are observed between the +3 basic residue and +1 Glu, which then form polar contacts with the ε_2_ NH group of the second specificity pocket His (+15). Similar to the role of the acid/histidine interaction of the catalytic triad [62], this chain might optimise the orientation and charge distribution at +15 His, which would otherwise be neutral at physiological pH [63]. This would encourage His to accept polar groups, such as the polar hydroxyl group of the ligand tyrosine, whose hydrophobic ring then contacts the conserved second specificity pocket aliphatic residue at +13, as described before [49]. This neutralises the role of +3 Lys in further ligand selectivity, whereas inclusion of +3 Arg maintains this interaction but allows for additional selectivity for acidic or phosphorylated residues C-terminal to the PPxY motif through its guanidino group, as described before [32]. The presence of Lys and not Arg at this position does not appear to prevent in vivo ubiquitination of Smad7 by WWP2-C [27], which contains only the WW4 and HECT domains. The binding mechanism in this instance might allow a low-affinity, low-specificity transient interaction that is sufficient to induce ubiquitination. As discussed above, SMURFs bind to Smad7 and translocate to the cell surface, a process that would necessitate high-affinity binding. This reduces TGFβ-induced transcriptional activity by reducing receptor activation of Smads, whereas WWP2-C increases TGFβ activity by degrading Smad7 to reduce the negative feedback. A weaker, transient interaction might suit this function in order to minimise ubiquitination of Smad7 binding partners.

Smad7 S206 was identified as a candidate phosphorylation site using prediction servers. This was corroborated by evidence in the literature that suggests this residue undergoes phosphorylation in vivo [47]. S206-phosphorylated Smad7 has an increased affinity for WWP2 WW4. Another Smad7 binding partner, YAP, supresses TGFβ signalling by potentiating Smad7 activity [64]. Observations from a YAP WW1/Smad7 co-structure suggest that phosphorylation at S206 would disrupt the binding mechanism [65]. Kinase pathway activity may, therefore, switch Smad7 activity from YAP-mediated suppression of TGFβ signalling to WWP2-mediated turnover, similar to that of other Smads [26,66]. This may explain the increased rate of Smad7 turnover we observed in vivo upon TGFβ stimulation [27], and such cross-talk would allow for contextualised cellular responses to TGFβ signalling inputs [33,67,68].

Evidence of a catalytically inactive, tandem WW3–WW4 domain-containing WWP2C-ΔHECT isoform suggests a further layer of regulation involving RNA splicing. Alternative splicing has been shown to be a key component in metastatic phenotypes [69], and TGFβ-mediated splice variation drives epithelial–mesenchymal transition [50], a commonly misregulated process in carcinomas. As in the case of WWP2 isoforms, these splicing programmes can regulate signalling cascades by altering E3 ligase activity. We observed that this putative isoform reduces activity at the Smad promoter. We cannot conclusively attribute these observations to the conservation of Smad7 inhibitory activity at this stage, especially since PPxY is a common motif conserved amongst other Smads and because WW domains are often promiscuous in their choice of binding partners. However, since our titration data showed a high WW3/Smad7 binding affinity when in tandem conformation, we propose that the WWP2C-ΔHECT isoform participates in protective binding of Smad7. This would potentiate TGFβ-suppressive activity by competing with catalytically active E3 ligases for access to the PPxY motif. In the context of metastatic cell lines, in which we found evidence of expression, prolonged inhibitory activity of Smad7 would allow the cell to escape cytostasis induced by the TGFβ gene programme. We found that WW4 holds the same conformation and has the same Smad7 binding mechanism and affinity both as a tandem domain and as a single domain. This might be a necessity for a WW domain that has to function in both scenarios in vivo as part of both WWP2-FL and WWP2-C. Our data suggest that as part of WWP2-FL, WW4 faces competition from its high-affinity neighbour, effectively reducing its role in ligand recruitment.

Our results put WWP2 at the junction of several regulatory networks involving ubiquitination, phosphorylation, splice variation and promoter activity. This suits the TGFβ signalling paradigm, where such diverse regulatory inputs allow contextualised responses that drive numerous physiological processes. The role that Smad7 PPxY motif phosphorylation and WWP2C-ΔHECT play in regulating these physiological processes and their role in disease states where contextualised TGFβ responses malfunction, will be interesting to study in the future. The WWP2 WW4 atomic structure elucidated here can be used as the foundation for further study on the rational design of therapeutics to target this disease-associated E3 ligase with small-molecule or peptidomimetic inhibitors. Indeed, coincidentally, we have already produced a modified Smad7 peptide with enhanced affinity in the form of pSmad7, although not systematic in our approach. The identification of binding-site residues makes an important contribution in this respect, and the binding hotspot residues 470Thr and 471Thr offer a starting point for targeted design, although the conservation of these residues across other WW domains poses a challenge when considering specificity. The role of WW3 and the tandem WW3–WW4 pair have revealed intriguing detail on the WWP2 structure–function relationship and re-emphasised the importance of tandem WW domains in substrate recruitment. Substrate-bound co-structures and mutational analysis will provide further valuable information on the interaction interface and mechanism of tandem-domain cooperation.

## 4. Materials and Methods

### 4.1. Protein and Peptide Expression and Purification

GB1-tagged constructs with an N-terminal hexa-His tag were generated by amplifying DNA fragments of single or tandem WWP2 WW domains and inserting them into the pSKDuet01 vector (Addgene plasmid #12172) using BamHI and HindIII restriction sites. SUMO-tagged constructs with an N-terminal hexa-his tag were generated by amplifying a Smad7 DNA fragment corresponding to Ser199 to Asp217 and inserting them into a modified pET21d(+) vector (a gift from Dr Robin Maytum, University of Bedfordshire) using BsaI and BamHI restriction sites. Single-^15^N isotope-labelled and double-^15^N, ^13^C isotope-labelled proteins were expressed in *E. coli* BL21 Star (DE3) (Fisher Scientific) using isotope-enriched M9 minimal medium supplemented with micronutrients and MEM vitamin solution (Sigma). Unlabelled proteins were expressed in LB medium using *E. coli* BL21 Star (DE3) (Fisher Scientific). Cells were harvested by centrifugation, and lysates were produced using a French Press at 10,000 psi. Recombinant proteins were purified using nickel-charged HisTrap FF columns (GE Healthcare) and the ÄKTA-FPLC system (GE Healthcare). ULP-1 (ubiquitin-like-specific protease-1) was used to cleave peptide His-SUMO tags overnight in Tris buffer. Protein His-tags were cleaved by thrombin digestion overnight in PBS. Tags were removed by a second round of nickel-affinity purification. Recombinant proteins were purified further by size-exclusion chromatography using the HiLoad 16/600 Superdex 75 prep grade column (GE Healthcare). Proteins were concentrated and prepared for NMR using 20 mM Na_2_HPO_4_, pH 6.8, 50 mM NaCl, 200 µM 4,4-dimethyl-4-silapentane-1-sulfonic acid, 10% D_2_O v/v and NaN_3_ 0.03% buffer. Protein concentrations were determined using extinction coefficients and absorbance at 280 nm. Synthetic peptides were purchased from Proteogenix, France.

### 4.2. NMR Spectroscopy

NMR data were collected at 298 K using either a Bruker Avance III 800 MHz spectrometer or a Bruker Avance I 500 MHz spectrometer. The spectra used for the backbone and sidechain assignment were: ^1^H-^15^N-HSQC, ^1^H-^13^C-HSQC, CBCA(CO)NH, HNCACB, CC(CO)NH, H(CCO)NH, H(C)CH-TOCSY, (H)CCH-TOCSY, hbCBcgcdceHE, hbCBcgcdHD, ^1^H-^13^C-TROSY-HSQC centered in the aromatic region, and a ^1^H-^1^H 2D plane of ^13^C-NOESY-HSQC with the ^13^C centered at 120 ppm (50 ms mixing time). Acquisition parameters can be found in Appendix A. Through-space distance restraints for the structural calculation were determined with ^15^N-NOESY–HSQC (100 ms mixing time) and ^13^C-NOESY–HSQC (100 ms mixing time) spectra. Spectra were processed using the NMRPipe software suite [70] and analysed using the CCPNmr Analysis software package [71].

### 4.3. NMR Ligand Titration

*Synthetic peptide:*^15^N-labelled protein was prepared at 0.78 mM, and 100 mM Smad7 or pSmad7 peptide was titrated in seven increments (1:0, 1:0.1, 1:0.2, 1:0.5, 1:1, 1:2, 1:4) to a ratio of 1:4. A ^1^H-^15^N-HSQC spectrum was taken at each titration point.

*Recombinant peptide:* Two samples of each ^15^N-labelled protein were prepared at 0.08 mM, one of which contained unlabelled Smad ligand at a ratio of 10:1. A titration was performed at 298 K by replacing aliquots of the non-ligand sample with aliquots of the ligand sample at increasing volumes, so that ligand concentration increased while protein concentration and sample volume remained the same. Ten titration increments were performed (1:0, 1:0.1, 1:0.2, 1:0.5, 1:1, 1:1.9, 1:3.6, 1:5.2, 1:6.8, 1:8.4). A ^1^H-^15^N-HSQC spectrum was taken at each titration point. 

### 4.4. Dissociation Constants

Changes in shift of amide peaks were weighted according to gyromagnetic ratio. For single-domain titrations, the following protein-ligand fast-exchange equation was fit to the saturation curves of each amide resonance using CcpNmr Analysis [71]. Residues with shift distances larger than or equal to the mean shift distance plus one standard deviation were averaged to calculate the K_D_.
$$y=A\left(B+x-\;\sqrt{{\left(B+x\right)}^{2}-4x}\right)$$a=total protein concentrationb=total ligand concentrationx=b/ay=change in observed chemical shift$$A=\;\Delta \delta_{\infty }/2$$$$B=1+K_{d}/a$$$$\Delta \delta_{\infty }=difference\;between\;start\;chemical\;shift\;and\;shift\;at\;saturation$$

For each tandem domain binding site, residues with shift distances larger than or equal to the mean shift distance plus one standard deviation were averaged, and these two means were summed. The xcrvfit [72] XY2 two-site binding equation [73] described below was fit against the binding curve.$$y=\left(\delta_{1}a+\frac{\delta_{2}a\left(x-a\right)}{2a+K_{2}}\right)÷\left[P\right]$$$$a=2\sqrt{\frac{-g}{3}}\times cos\left(\left(240+\frac{h}{3}\right)\times \frac{\pi }{180}\right)-\frac{d}{3}$$$$b=K_{2}-4K_{1}$$$$c=K_{2}K_{2}$$$$d=\frac{\left(c-4K_{1}K_{2}\right)-2\left[P\right]K_{2}}{b}$$$$e=\frac{xK_{2}\left(2\left[P\right]-x\right)-c\left(\left[P\right]+K_{1}+x\right)}{b}$$$$f=\frac{-\left(2d^{3}-9de+27\left(cx\left[P\right]÷b\right)\right)}{54}$$$$g=\frac{3e-d^{2}}{3}$$$$h=\frac{180}{\pi \left(cos^{-1}\left(w\right)\right)}$$$$w=\frac{f}{\sqrt{\left(-g^{3}÷27\right)}}$$x=ligand concentrationy=change in observed chemical shift$$K_{1}=K_{D}\;binding\;site\;1$$$$K_{2}=K_{D}\;binding\;site\;2$$$$\delta_{1}=difference\;between\;binding\;site\;1\;start\;chemical\;shift\;and\;shift\;at\;saturation$$$$\delta_{2}=difference\;between\;binding\;site\;2\;start\;chemical\;shift\;and\;shift\;at\;saturation$$[P]=protein concentration


### 4.5. Structure Determination

Unassigned ^13^C-NOESY–HSQC and ^15^N-NOESY–HSQC distance restraints were calculated with CCPNmr Analysis [71], and φ and ψ dihedral restraints were calculated using DANGLE [74]. Both sets of restraints, the ^13^C-NOESY-HSQC and ^15^N-NOESY-HSQC spectra, were used to calculate the 20-model refined protein structure ensemble using ARIA software embedded in the CCPN grid and the WeNMR ARIA 1.2 HJ webportal [75,76,77,78]. The ensemble was validated using PSVS 1.5 [79], Molprobity [80] and wwPDB [81]. The coordinates were deposited in the Protein Data Bank under accession code 6RSS. RMSD values were generated for well-defined regions (construct residues 3–59 for GB1 and 75–101 for WW4), as determined by wwPDB.

### 4.6. Cytokine Stimulation and Immunoblotting

HEK-293 cells were transfected with mouse or human ESRP homologue expression plasmids as described previously [82]. Cells were stimulated with 5 ng/mL TGFβ for 1 h, washed in ice-cold PBS, and lysed in 1% v/v Igepal-630, 50 mM Tris pH 8.0, 150 mM NaCl, 10% v/v glycerol, 5 mM EDTA, 1 mM NaF, 1 mM Na_3_VO_4_ and protease inhibitors (1% NP-40 LB), in the ratio 1 tablet (Roche):30 mL 1% NP-40LB. Clarified lysates were boiled in Laemlli buffer with DTT and analysed by western blot using anti-WWP2 antibody specific for a C-terminal epitope (Santa Cruz Biotechnology).

### 4.7. Semi-Quantitative RT-PCR

RNA was extracted from TGFβ-stimulated A375, COLO357, SK-MEL28 and VCaP cells at increasing time intervals using the SV Total RNA Isolation kit (Promega). Reverse transcription was performed using random primers (Promega) and GoScript reverse transcriptase (Promega). WWP2-ΔHECT specific primers (Forward: 5’-GCTGGGAAGAACAATTACTG-3’ Reverse: 5’-TTCCTCTGTAACATGCTCCCT-3’) were used in conjunction with the GoScript kit (Promega), and the DNA products were analysed using 1% agarose-TAE gel.

### 4.8. Luciferase Reporter Assays

The Smad3 reporter construct (pCAGAC_12_-luc) was kindly provided by Caroline Hill (CRUK Laboratories, Francis Crick Institute, UK). For each plasmid transfection, 250 ng pCAGAC_12_-luc- and 15 ng pRSV-β-galactosidase (pRSV-βgal)-encoding plasmid were used in conjunction with Smad-encoding plasmids, as described previously [27]. Values were averaged, luciferase activity standardised for β-galactosidase activity, and data expressed as relative fold changes in luciferase activity over basal activity.

## Figures and Tables

**Figure 1 ijms-20-04682-f001:**
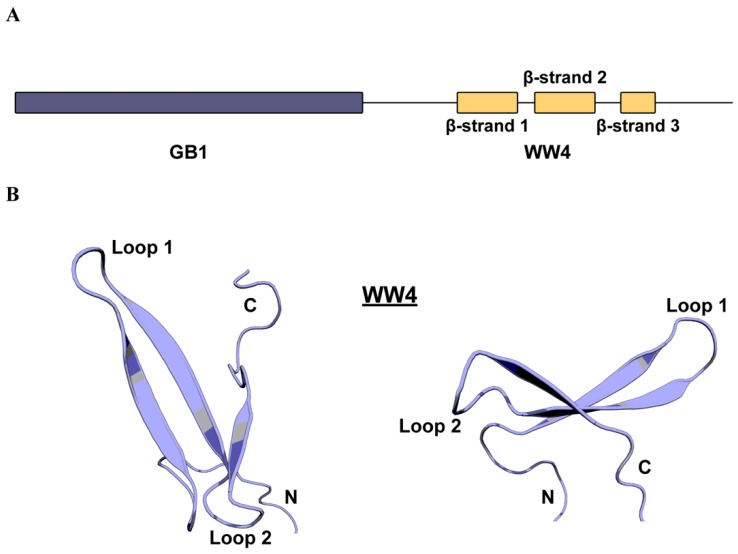
The three-stranded anti-parallel β-sheet structure of WWP2 WW4 solved by NMR. (**A**) A schematic of the B1 domain of streptococcal protein G (GB1)–WW4 recombinant protein. (**B**) Ribbon diagram depictions of the WWP2 WW4 domain from the most representative model (model 1) of the refined 20-model structural ensemble.

**Figure 2 ijms-20-04682-f002:**
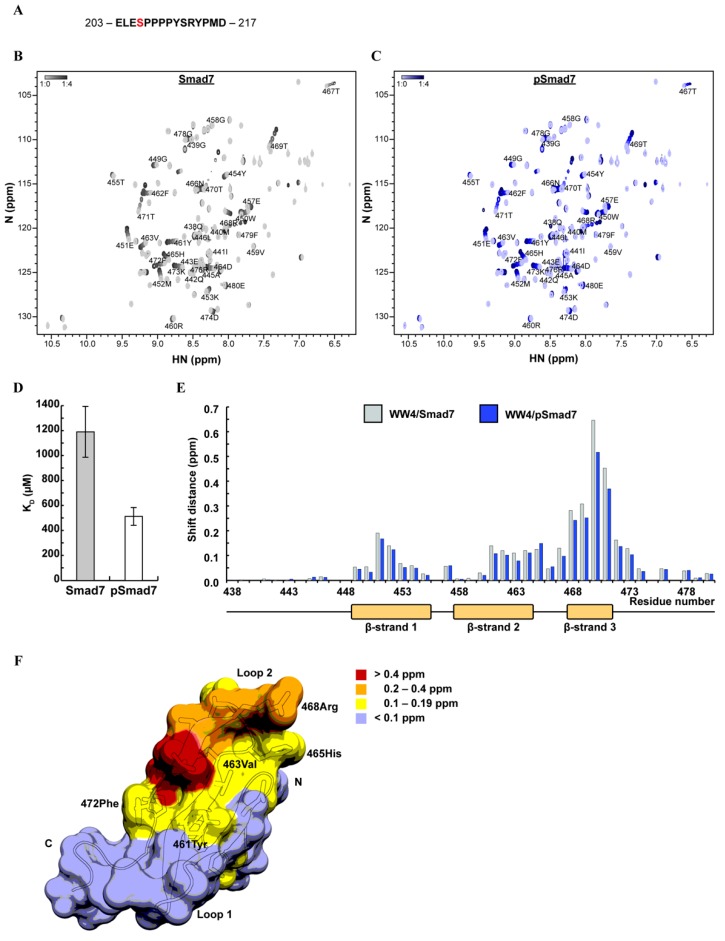
The interaction between WWP2 WW4 and Smad7/S206-phosphorylated Smad7 (pSmad7). (**A**) The Smad7 ligand, with the PPxY motif in bold and the putative S206 phosphorylation site in red. (**B**) The superimposed WW4/Smad7 titration heteronuclear single quantum coherence (HSQCs). Lower ligand concentrations are in light grey, and higher ligand concentrations are in dark grey. (**C**) The superimposed WW4/pSmad7 titration HSQCs. Lower ligand concentrations are in light blue, and higher ligand concentrations are in dark blue. (**D**) The WW4/Smad7 (1.19 ± 0.21 mM) and WW4/pSmad7 (0.5 ± 0.07 mM) dissociation constants. (**E**) The shift distances of the WW4/Smad7 and WW4/pSmad7 titration HSQC amide peaks in ppm. The WW4 secondary structure is aligned to the residue number along the x-axis, with β-strands represented as orange boxes. (**F**) The WWP2 WW4 domain structure with Smad7 binding site residues colour-coded for titration shift distances. The labelled residues are the XP-binding groove (472Phe and 461Tyr) and the second specificity pocket (463Val, 465His and 468Arg), which determine specificity for the PPxY recognition motif.

**Figure 3 ijms-20-04682-f003:**
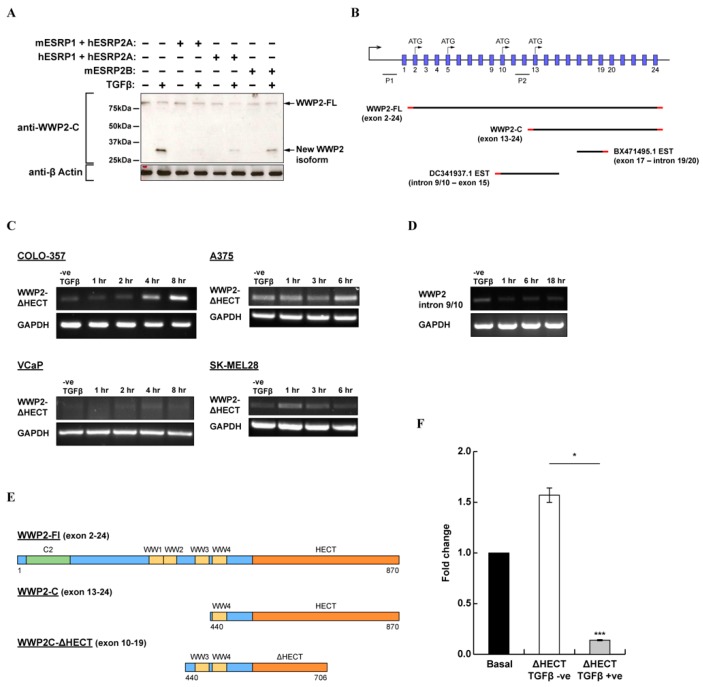
Splice variation at the *wwp2* gene. (**A**) A Western blot of HEK293A cells transfected with combinations of mouse and/or human epithelial splicing regulatory proteins (ESRPs) as indicated and stimulated with TGFβ; the membrane was probed with anti-WWP2C antibody or anti-β-actin antibody as a control. (**B**) The *wwp2* gene locus (not to scale) with aligned transcripts and expressed sequence tag (ESTs) as annotated. (**C**) Semi-quantitative RT-PCR using primers targeting intron 9/10 of *wwp2* mRNA extracted from A375 cells treated with TGFβ over 18 h. (**D**) Semi-quantitative RT-PCR using primers targeting exon 17–intron 19/20 of *wwp2* mRNA extracted from COLO-357 and VCaP cells treated with TGFβ over 8 h and from A375 and SK-MEL28 cells treated over 6 h. (**E**) A schematic showing the domain composition of WWP2-FL, WWP2-C and the putative new isoform WWP2C-ΔHECT. (**F**) Fold change in luciferase activity in HEK293A cells co-transfected with WWP2C-ΔHECT and the Smad3-dependent CAGA12-luciferase reporter, treated and untreated with TGFβ, with standard error bars.

**Figure 4 ijms-20-04682-f004:**
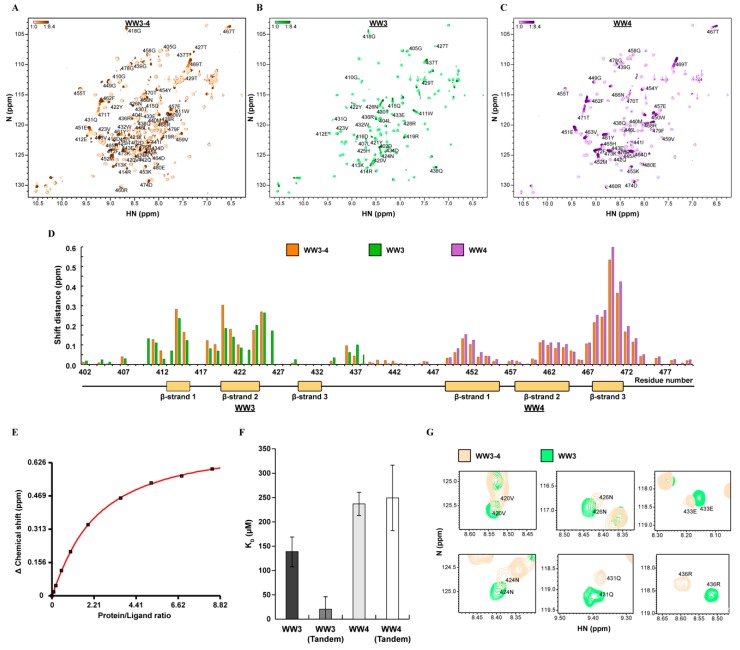
The interaction between tandem WWP2 WW3–4 and Smad7. (**A**) The superimposed WW3–4/Smad7 titration HSQCs. Lower ligand concentrations are in light orange, and higher ligand concentrations are in dark orange. (**B**) The superimposed WW3/Smad7 titration HSQCs. Lower ligand concentrations are in light green, and higher ligand concentrations are in dark green. (**C**) The superimposed WW4/Smad7 titration HSQCs. Lower ligand concentrations are in light purple, and higher ligand concentrations are in dark purple. (**D**) The shift distances of the WW3–4/Smad7 (orange), WW3/Smad7 (green) and WW4/Smad7 (purple) titration HSQC amide peaks in ppm. The WW3–4 secondary structure is aligned to the residue number along the x-axis, with β-strands represented as yellow boxes. (**E**) The WW3–4/Smad7 binding curve K_D_ fit. (**F**) The WW3/Smad7 (139 ± 14.4 µM), WW3(tandem)/Smad7 (20.57 ± 26.24 µM), WW4/Smad7 (237 ± 15.7 µM) and WW4(tandem)/Smad7 (249.3 ± 68.1 µM) dissociation constants. (**G**) HSQC peaks belonging to the WW3 domain at different chemical shifts for WW3–4 (orange) and WW3 (green).

**Table 1 ijms-20-04682-t001:** Summary of the NMR data and structure validation statistics for each domain.

**NMR Distance and Dihedral Constraints**	**GB1–WW4**	
**Distance constraints**		
Total NOE	1803	
Intra-residue	873	
Inter-residue		
Sequential (|i-j| = 1)	400	
Medium-range (|i-j| < 4)	118	
Long-range (|i-j| > 5)	412	
Total dihedral angle restraints		
φ	96	
ψ	96	
**Structure statistics**	**GB1**	**WW4**
Violations (mean and s.d.)		
Distance constraints (±0.3Å tolerance)	2	0
Dihedral angle constraints (±5˚ tolerance)	2	1
Max. dihedral angle violation (˚)	1.75	2.24
Max. distance constraint violation (Å)	0.043	-
Deviations from idealised geometry		
Bond lengths (Å)	0.008	0.009
Bond angles (˚)	0.9	0.8
Impropers (˚)	-	-
Average pairwise RMSD ^1^ (20 refined structures) (Å)		
Heavy	0.8	0.9
Backbone	0.4	0.4

^1^ Root-mean-square deviation (RMSD) for well-defined regions (GB1 residues 3–59, WW4 residues 75–101). Nuclear Overhauser effect (NOE).

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
