# Peer review of "Smad7 Binds Differently to Individual and Tandem WW3 and WW4 Domains of WWP2 Ubiquitin Ligase Isoforms"

_ijms, 2019, doi:10.3390/ijms20194682_

Round 1
Reviewer 1 Report
The manuscript is well written and the aims and conclusions are consistent. The work adds valuable information regarding WW2P proteins and also regarding WW domains. This information includes the description of a new isoform and a new WW domain structure. It also describes that WW domain pairs bind with higher affinity than individual domains to the same ligand, highlighting the relevance of studying domain pairs -and not only isolated domains- when binding modes are characterized are when molecules used as inhibitory targets are designed.
There is abundant information in the literature describing examples of WW domain aggregation. It is a pity that this propensity precluded the determination of the complex structures of these domains bound to the Smad7 peptides, (with and without phosphorylated serine). However, the authors succeeded in mapping the binding area on the domain and this information can be very valuable for future experiments. Perhaps, if the authors continue with this research, they could test the addition of the peptide during purification, before removing the fusion protein. Perhaps the WW-peptide complex is more soluble than the isolated domain.
Some minor comments are listed below.
The figures describing the HSQC experiments are too small and is difficult to read the labels in the printed version of the manuscript. Perhaps the authors can provide a large version of these panels as supplementary information.
Figure S3 contains some errors regarding the SMAD DNA binding sites. I have included some modifications in the original Figure S3 to indicate the mistakes and the correct Smad binding elements. I have also indicated some of these Smad binding sites in yellow but there are many more that I did not label. Please add these sites to the revised version of this figure and remove the CAGA sites (this is a name for the sites, which are AGAC and 5GC sites).
Please see a description of these sites in the manuscript
Martin-Malpartida P, et al. (2017) Structural basis for genome wide recognition of 5-bp GC motifs by SMAD transcription factors. Nat Commun 8(1):2070

Author Response
Response to Reviewer 1
Thank you for taking the time to review our manuscript and for your kind comments. We have noted your suggested revisions and have modified the manuscript as follows;
Point 1: The figures describing the HSQC experiments are too small and is difficult to read the labels in the printed version of the manuscript. Perhaps the authors can provide a large version of these panels as supplementary information.
Response 1: Large versions of the HSQC spectra have now been included in the supplementary section so that the content of these Figures is legible when printed.
Point 2: Figure S3 contains some errors regarding the SMAD DNA binding sites. I have included some modifications in the original Figure S3 to indicate the mistakes and the correct Smad binding elements. I have also indicated some of these Smad binding sites in yellow but there are many more that I did not label. Please add these sites to the revised version of this figure and remove the CAGA sites (this is a name for the sites, which are AGAC and 5GC sites).
Response 2: The CAGA sites have been removed and the following sites have been included in accordance with the findings in the cited paper:
SBE sites: GTCT and GTCTG
5GC sites: GGCT, GGCTG, GGCGC and GGCCG
Reviewer 2 Report
Overall, this is a nicely presented paper with significant implications. The work is well done and appropriately interpreted. however, I do not believe the authors adequately identified SMAD7 as an antagonist of TGF-beta signaling. The physiologic and pathophysiologic context of the importance of their findings are largely left up to the reader to determine. This needs to be addressed. Moreover, it is not immediately clear is the data in this paper refer uniquely to SMAD7 or apply to other TGF-beta pathway negative regulators. These possibilities are bioinformatically accessible and should at least be discussed to understand the broader ramifications (if any) of the findings presented in this paper.
Author Response
Response to Reviewer 2
Thank you for taking the time to review our manuscript and for your kind comments. We have noted your suggested revisions and have modified the manuscript as follows;
Point 1: Overall, this is a nicely presented paper with significant implications. The work is well done and appropriately interpreted. however, I do not believe the authors adequately identified SMAD7 as an antagonist of TGF-beta signaling.
Response 1: The TGFβ signalling pathway has been explained in more detail in the introductory section to clarify any ambiguity around the role of Smad7 as an antagonist.
Point 2: The physiologic and pathophysiologic context of the importance of their findings are largely left up to the reader to determine.
Response 2: The physiological and pathophysiological role of TGFβ signalling has been explained in more detail in the introductory section in order to provide context to the importance of the findings from this work. Reference has also been made to this in the discussion section.
Point 3: Moreover, it is not immediately clear is the data in this paper refer uniquely to SMAD7 or apply to other TGF-beta pathway negative regulators. These possibilities are bioinformatically accessible and should at least be discussed to understand the broader ramifications (if any) of the findings presented in this paper.
Response 3: We have added some discussion to describe the potential for other PPxY motif-containing proteins to produce the TGFβ-inhibitory activity observed by WWP2C‑ΔHECT, and have also proposed a potential “low specificity” binding mechanism for the WW4 domain.